# Achieving Cycling Stability in Anode of Lithium-Ion Batteries with Silicon-Embedded Titanium Oxynitride Microsphere

**DOI:** 10.3390/nano13010132

**Published:** 2022-12-27

**Authors:** Sung Eun Wang, DoHoon Kim, Min Ji Kim, Jung Hyun Kim, Yun Chan Kang, Kwang Chul Roh, Junghyun Choi, Hyung Woo Lee, Dae Soo Jung

**Affiliations:** 1Energy Storage Materials Center, Korea Institute of Ceramic Engineering & Technology (KICET), Jinju-si 52851, Republic of Korea; 2Department of Nanoenergy Engineering, Pusan National University, Pusan 46241, Republic of Korea; 3Department of Materials Science and Engineering, Korea University, Seoul 02841, Republic of Korea; 4Department of Nano Fusion Engineering and Research Center of Energy Convergence Technology, Pusan University, Pusan 46241, Republic of Korea

**Keywords:** lithium-ion batteries, silicon anodes, titanium oxynitrides, spray-drying

## Abstract

Surface coating approaches for silicon (Si) have demonstrated potential for use as anodes in lithium-ion batteries (LIBs) to address the large volume change and low conductivity of Si. However, the practical application of these approaches remains a challenge because they do not effectively accommodate the pulverization of Si during cycling or require complex processes. Herein, Si-embedded titanium oxynitride (Si-TiON) was proposed and successfully fabricated using a spray-drying process. TiON can be uniformly coated on the Si surface via self-assembly, which can enhance the Si utilization and electrode stability. This is because TiON exhibits high mechanical strength and electrical conductivity, allowing it to act as a rigid and electrically conductive matrix. As a result, the Si-TiON electrodes delivered an initial reversible capacity of 1663 mA h g^−1^ with remarkably enhanced capacity retention and rate performance.

## 1. Introduction

To meet the growing demand in the fields of electronic vehicles and energy storage systems, developing high-capacity electrode materials is crucial for enhancing the energy density of lithium-ion batteries (LIBs) [1,2,3,4]. Silicon (Si) has been considered as a promising anode material with an outstanding theoretical capacity (4200 mA h g^−1^, Li_22_Si_5_), which is 10 times larger than the traditional graphite anode (372 mA h g^−1^, LiC_6_) [5]. However, commercialization of Si anodes is still hampered by its volume expansion problem, resulting in severe capacity fading during lithiation/delithiation. These shortcomings are exacerbated by the pulverization of particles and the repeated formation of solid-electrolyte-interphase (SEI), which leads to continuous electrolyte consumption during cycling [6,7,8,9,10,11,12,13].

Generally, the simple coating on the Si surface using various functional materials is widely used to mitigate electrochemical and mechanical degradation [14,15,16,17,18,19,20,21,22]. The coating layer can help avoid direct contact between Si and the electrolyte, reducing the growth of the unstable SEI, while it also acts as a volume expansion buffer to alleviate fracture of the electrode [23,24]. Carbonaceous materials are the commonly used candidates for the surface modification of Si due to their high electrical conductivity and chemical stability with electrolytes. However, most of them are rigid, resulting in poor durability against Si volume expansion. Other inorganic species, including metal oxides (e.g., TiO_2_, MgO, and Al_2_O_3_) and SiO_x_ have also been explored, which can enhance the structural integrity with good mechanical properties and act as artificial SEI layers to suppress side reactions during cycling [25,26]. Nevertheless, the poor electrical conductivity and lithium loss at the first cycle may cause insufficient electrochemical performances. 

Herein, we propose titanium oxynitride-embedded Si NPs (Si-TiON) achieving high cycling stability in LIBs and successfully fabricated via a spray-drying process. TiO_2_ can be uniformly and thinly coated on the Si surface during the drying process, and then converted into TiON under the optimized nitridation conditions. TiON exhibits high conductivity and mechanical strength, which facilitates electron/ion migration and enhances the structural integrity of Si-based anodes [27,28,29]. Additionally, spray-drying used for Si-TiON manufacturing plays a key role in the formation of the core-shell structure via capillary force and is widely applied to industrial fields due to its scalable and low-cost properties [30,31,32]. By virtue of these properties, the Si-TiON composite can be a promising candidate for the practical use of high-performance Si-based anodes.

## 2. Materials and Methods

### 2.1. Synthesis of TiON as a New Coating Material

To synthesize optimized TiON, TiO_2_ nanoparticles (TiO_2_ NPs, average diameter = 20 nm, Degussa P25 (anatase 85%, rutile 15% phase), Evonik, Essen, Germany) were nitrided at different temperatures (700 °C, 750 °C, 800 °C, 850 °C, 900 °C, and 950 °C) under ammonia gas (NH_3_, 99.9%, flow rate: 1.5 L/min) for 5 h in a quartz tube furnace.

### 2.2. Preparation of Si-TiON Composite

One g of Si nanoparticles (Si NPs, ~50 nm, 99.9%, Nano & Amor Inc., Houston, TX, USA) and 1 g of polyvinyl pyrrolidone (PVP, Mw. 200,000, Sigma-Aldrich Co., St. Louis, MO, USA) were dispersed in 200 mL of deionized (DI) water and stirred for 30 min. Then, 1 g of TiO_2_ NPs was added and agitated for 30 min using a high-energy tip sonicator (750 W, VCX 750, Sonics & Materials Inc., Newtown, CT, USA) to obtain a homogeneous solution. To synthesize Si-TiON composite powders, the precursor was loaded into a spray dryer (B-290, BUCHI Labortechnik AG, Flawil, Switzerland). The detailed parameters were as follows: inlet temperature—180 °C, feeding speed—5 mL min^−1^, and air flow rate—30 L min^−1^. After drying, the prepared Si-TiO_2_ composite underwent thermal treatment at 800 °C for 5 h under an atmosphere of ammonia (NH_3_, 99.9%) and ramping speed maintained at 5 °C min^−1^. To optimize the amount of TiON, TiO_2_ was added in different ratios in the precursor. The amount of TiO_2_ corresponding to 1 g of Si was 0.5 g, 1 g, and 2 g, and denoted as “Si-TiON (1:0.5)”, “Si-TiON (1:1)”, and “Si-TiON (1:2)”, respectively.

### 2.3. Material Characterization

The structures of Si-TiO_2_ and Si-TiON were characterized by X-ray diffraction (XRD, D8 Advance, Bruker, Billerica, MA, USA) using Cu Kα radiation. Nitrogen adsorption and d,esorption isotherms were obtained with a 3-Flex (Micromeritics, Norcross, GA, USA), and the specific surface area was obtained using the Brunauer-Emmett-Teller (BET) method. The chemical binding was observed by X-ray photoelectron spectroscopy (XPS, VersaProbe, Ulvac-PHI Inc., Yokohama, Japan). The morphology investigations were performed using a field emission scanning electron microscope (FE-SEM, JSM-7610F, JEOL, Tokyo, Japan), and high-resolution transmission electron microscopy (HR-TEM, Themis Z, FEI Co., Hillsboro, OG, USA). For the measurement of conductivity of synthesized TiON, a current-voltage (I-V) source was used. Teflon mold was filled with the powder and loaded under the probe of a 20 mm diameter electrode. The conductivities of the TiON were obtained from the gradient of the I-V curves.

### 2.4. Electrochemical Measurements

The mixture of Si-TiON composite (60 wt%), super-P (20 wt%), and poly-acrylic acid (20 wt%) was stirred to form a homogeneous slurry. The electrode was prepared on Cu foil (18 μm, Hohsen Corp., Tokyo, Japan) using a doctor blade method. Then, the electrode was dried at 80 °C overnight under vacuum and punched into circular discs for coin-cell fabrication. The Si NPs electrode was also prepared separately under the same conditions. The 2032-type coin half-cell was assembled in a dry room. The electrolyte was 1M LiPF_6_ in a mixture of ethylene carbonate (EC) and diethyl carbonate (DEC) (1:1 v/v%) with 5 wt% fluoroethylene carbonate (FEC, Dongwha Electrolyte Co., LTD., Nonsan-si, Korea). The cell was charged and discharged galvanostatically in the range of 0.01–1.5 V on a Won-A-Tech battery tester.

## 3. Results and Discussion

TiON was prepared by thermal treatment of TiO_2_ under a NH_3_ atmosphere at different temperatures. The phase transformation of TiO_2_ after nitridation was observed by X-ray diffraction (XRD) patterns shown in Figure 1a. The TiO_2_ NPs (Degussa P25) used as the starting material for the preparation of TiON are composed of mainly the anatase phase (85%) and partially the rutile phase (15%), as is well-known [33]. At a low nitridation temperature at 700 °C, the major peak of the rock-salt TiON phase appeared at 43.2° with the peaks of the anatase and rutile TiO_2_ phases being at 25.2° and 27.4°, respectively [34]. As the temperature increased, the peaks corresponding to TiO_2_ were no longer detected, while those of TiON drastically increased. These results indicate that TiO_2_ is completely phase-transformed into TiON at a temperature above 750 °C. Figure 1b shows the magnified XRD patterns for the main peak of TiON in the range from 42.0 to 44.0°. TiON is a solid solution of TiO and TiN, and a diffraction peak of TiON appears between the peaks corresponding to these two phases. It can be observed that the diffraction peaks shift to a lower degree corresponding to the TiN region at higher nitridation temperatures. It demonstrates that more anion exchanges of O and N atoms occurred at a high temperature, resulting in a TiN-rich phase. A formation mechanism of TiON is proposed as follows. The TiO_2_ phase is converted into the TiO phase under an atmosphere of NH_3_ with high reduction ability. The TiO phase contains large oxygen vacancies (~15%) owing to its large non-stoichiometry. The oxygen vacancy is a thermodynamically stable site for incorporating N atoms, and Ti^2+^ offers electrons to N atoms for the stabilization of N^3−^ [35,36]. This could facilitate the exchange of O and N atoms resulting in the formation of TiON phase. The electrical conductivity of TiON powder was compared with that of super-P commonly used as a conductor in LIBs, as shown in Figure 1c. TiON nitrided at 800 °C shows higher electrical conductivity than TiON nitrided at 700 °C in the whole range of pressure. TiON can reach a high electrical conductivity of 3.60 S cm^−1^, which is similar to that of super-P (3.68 S cm^−1^) at a high pressure of 150 kgf cm^−2^. This result shows that the TiN phase formed at a high nitridation temperature can enhance electrical conductivity and act as a conductive agent for Si. Figure 1d shows the N_2_ adsorption/desorption isotherm and surface area of TiON. The plots almost overlapped at the nitridation temperatures of 700, 750, and 800 °C, indicating negligible change in surface area, but the surface area significantly decreased at temperatures above 850 °C. This is caused by the agglomeration of TiON particles owing to the metallic property of TiN manifested at high temperatures. Therefore, in this study, the nitridation temperature was fixed at 800 °C for the synthesis of Si-TiON composite to achieve high electrical conductivity without particle agglomeration.

A spray-drying process was used for the facile synthesis of Si-TiON composite, which is illustrated in Figure 2a. To prepare a precursor solution, Si NPs (50 nm), TiO_2_ (20 nm), and PVP were dispersed in DI water. First, the precursor droplets generated from the spray nozzle were injected into the reactor via drying gas. The droplet underwent solvent evaporation and drying to form micro-sized secondary particles based on the one droplet–one particle conversion [37,38]. During the drying process, the smaller TiO_2_ particles can move easily, while the relatively larger Si NPs move slowly in the droplet. This is because the Brownian motion effect is greater on smaller particles, resulting in faster capillary flow of TiO_2_ colloidal particles to the interface region where solvent evaporation occurs first. Thus, TiO_2_ particles were trapped in this region and mainly located at the shell of the particle, leaving the Si NPs in the core. Furthermore, the spaces between the Si NPs can be filled by the TiO_2_ particles due to the interparticle capillary force. This self-assembly of colloidal particles leads to the uniform embedding of Si NPs in TiO_2_ [39,40]. Subsequently, the Si-TiON composite can be obtained by the nitridation process. Figure 2b–d shows the morphology of the Si-TiON composite with different TiON contents. All the secondary particles present a spherical morphology with a particle size of about 5 μm. The high-magnification image in the inset of Figure 2b shows that the surface is composed of irregular nanoparticles with a size of 20–50 nm, indicating the coexistence of both TiON and Si NPs. On the other hand, as the TiON content increases, only small particles below 20 nm can be observed at the surface, as shown in the inset of Figure 2c,d. It can be concluded that Si NPs can be completely embedded in TiON if the ratio of TiON to Si exceeds 1:1.

The Si-TiON (1:1) composite was further characterized using transmission electron microscopy (TEM). Figure 3a,b shows that the secondary particles are formed of numerous nanoparticles and the interconnected shell region. The densely packed microsphere is a result of the flow of TiO_2_ particles between Si particles, as mentioned above. It is noteworthy that TiON formed from TiO_2_ is connected to each other due to the metallic property of TiN, which can further enhance the mechanical stability of coating layers against the volume expansion of Si. The high-resolution TEM images in Figure 3c indicate clear lattice fringes with *d*-spacings of 0.147 and 0.157 nm corresponding to rock-salt TiON in the shell (point 1) and crystalline Si (point 2) in the core, respectively [41,42]. In addition, energy-dispersive X-ray spectra (EDS) from point 1 shown in Figure 3d show a higher proportion of Ti than Si, while the opposite tendency can be observed for point 2, which decisively demonstrates that Si NPs were well-enclosed by the TiON particles. The elemental mapping images for Si, Ti, O, and N (Figure 3e) also indicate that Si NPs are dispersed in the TiON matrix, demonstrating a core-shell structure.

Figure 4a,b shows the XRD patterns of the Si-TiO_2_ and Si-TiON composites prepared with different ratios of TiO_2_. The typical diffraction peaks of the Si and TiO_2_ phases are revealed in the Si-TiO_2_ composite, but the peaks corresponding to TiO_2_ disappeared and peaks of TiON appeared in the Si-TiON composite. These indicate that the TiO_2_ phase was completely converted into the TiON phase after the subsequent nitridation. With an increase in the proportion of TiO_2_ in the Si-TiO_2_, the intensity of the peak corresponding to TiO_2_ increases, as shown in Figure 4a. It can be found that Si-TiON also shows the same tendency, which means that the coating material content increases in the composite. No other peaks except those corresponding to Si and TiON are observed because no side reactions occurred during the spray-drying and nitridation process. 

The XPS survey spectra also suggest the presence of TiON coating layers on the surface of Si (Figure 4c,d). The Ti 2p peaks show three oxidation states of Ti^4+^, Ti^3+^, and Ti^2+^ at 457.9, 456.6, and 455.1 eV, respectively [43,44,45]. The oxidation state of Ti^4+^ is related to the existence of TiO_2_ because TiON can be easily oxidized in the air. The Ti^2+^ component derives from the TiO phase, whereas the Ti^3+^ species come from the TiN phase, demonstrating that TiON is the solid solution of TiO and TiN. The N1s spectra can be deconvoluted into two major peaks corresponding to Ti–N (397.4 eV) and Ti–O–N bonding (396.3 eV), which is consistent with the results of Figure 4c [46,47]. N–O (398.5 eV) can be observed when the sample is oxidized by exposure to air. 

The electrochemical properties of Si-TiON composite and Si NPs were examined using coin-type half-cells with the Li metals as counter electrodes (Figure 5). In galvanostatic tests, Si NPs electrode delivered an initial specific capacity of 3193 mA h g^-1^ at a current density of 0.1 A g^−1^. On the other hand, Si-TiON (1:0.5) and (1:1) exhibited an initial reversible capacity of 2378 and 1663 mA h g^−1^, respectively. It can be found that Si-TiON (1:1) composite has a similar initial Coulombic efficiency of 80.1% compared to that of Si NPs (80.9%), which demonstrates that TiON is inactive towards lithium ions, preserving the reversibility of Si. At the subsequent cycles at 2 A g^−1^, bare Si NPs show severe capacity fading due to the absence of a buffer layer; the Si-TiON (1:0.5) composite also exhibits relatively low cycle stability because Si cannot be completely encapsulated due to the insufficient amount of TiON, making it difficult to suppress the volume expansion of Si (Figure 5b). However, the Si-TiON (1:1) composite achieved high capacity retention up to 30 cycles because the TiON coating layer was completely deposited on the surface of Si, improving the structural stability of the overall electrode. Si-TiON (1:2) also exhibits excellent initial Coulombic efficiency and cycle life, demonstrating the superiority of TiON as a coating material; however, it shows low reversible capacity due to the decrease in the total active material ratio (Appendix A). Appendix A shows that Si-TiO_2_ (1:1) delivered an initial reversible capacity of 1517 mA h g^−1^ with the Coulombic efficiency of 75.7%, which is smaller than that of Si-TiON (1:1) because of the irreversible reaction between TiO_2_ and lithium ions. To further prove the structural stability of Si-TiON (1:1), cross-sectional FE-SEM images after cycling were observed (Appendix A). A Si NP electrode shows severe electrode swelling and collapse due to the large volume expansion of active material, whereas the Si-TiON electrode indicates moderate swelling behaviors and maintains its integrity. This result shows that the TiON coating layer can enhance the stability of the electrode during cycling due to its outstanding mechanical properties.

Figure 6a shows the cycling performance and Columbic efficiency of the Si-TiON (1:1) composite. Samples were cycled at 1 A g^−1^ after pre-cycling at 0.1 A g^−1^ to form stable SEI. The initial reversible capacity of Si-TiON (1:1) was 1140 mA h g^−1^ with a capacity retention of 97% after 100 cycles, demonstrating the effect of the mechanical aspect of applying the TiON coating. Impedance spectroscopy (EIS) also exhibits the outstanding electrochemical properties of Si-TiON (1:1) (Appendix A). Si-TiON (1:1) shows a much smaller semicircle than Si NPs in the high-frequency region, which corresponds to the charge transfer resistance at the electrode/electrolyte interface. This implies that Si-TiON (1:1) has high structural stability due to the existence of the rigid coating layer, while Si NPs went through structural failure, resulting in the formation of a thick SEI layer during cycling. The rate performances of Si NPs and Si-TiON (1:1) under different current densities were compared in Figure 6b. As the current density increases gradually, the Si-TiON composite achieves superior rate capabilities, as seen from the measurements after every 10 cycles. The specific discharge capacities were determined to be 1411, 1249, 1102, 892, and 603 mA h g^−1^ at various current rates from 0.1 to 5 A g^−1^. In contrast, Si NPs show rapid capacity degradation for the first 30 cycles, and even inferior rate capabilities under higher current densities. This finding further demonstrates that the TiON shell layer can improve the electrical conductivity of the Si-based anode by facilitating electron and ion mobility, leading to enhanced rate performance [48]. The electrochemical performances of Si-TiON (1:1) fabricated from facile spray pyrolysis were compared to the various Si composites reported in the previous works (Appendix A); Si-TiON (1:1) shows high initial capacity, enhanced cycle life, and excellent rate capability compared to those of other Si-based anodes. These electrochemical performances are attributed to the TiON with excellent electrical and mechanical properties and unique Si-TiON structure in which electronic conductivity is supported by the TiON matrix through direct contact with Si NPs, while the structure accommodates the volume expansion of Si NPs.

## 4. Conclusions

In summary, the Si-TiON composite with a TiON coating material has been introduced and completely synthesized via a simple spray-drying. The final product shows that Si NPs are uniformly embedded in the TiON via self-assembly of colloidal particles during the drying process. The TiON component plays an important role in enhancing cycling stability and rate capability because it can alleviate the volume changes of Si NPs to maintain structural stability and support the electrical conductivity of the electrode. As a result, the Si-TiON composite exhibited high initial Coulombic efficiency, as well as excellent cycling performance. Furthermore, it showed a remarkably high rate capability in comparison with commercial Si NPs. This study offers a promising TiON coating material for the development of high-performance Si anodes for LIBs, and a simple fabrication method for the core-shell composite based on a scalable synthetic process.

## Figures and Tables

**Figure 1 nanomaterials-13-00132-f001:**
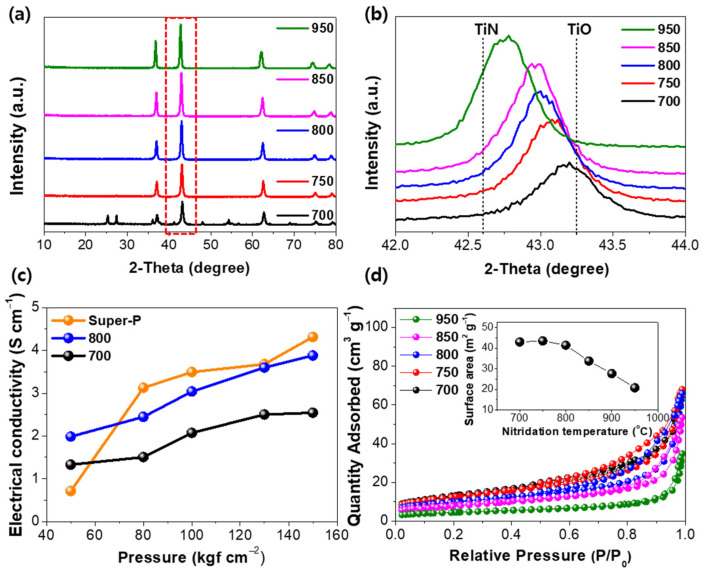
Physicochemical properties of TiON prepared with different nitridation temperatures. (**a**) XRD patterns and (**b**) magnified major peaks at 2-theta 42–44°. (**c**) Electrical conductivity behavior of TiON prepared at different nitridation temperatures as a function of pressure with those of a commercial carbon conductor (super-P). (**d**) Nitrogen adsorption and desorption isotherms; (Inset) surface area depends on the nitridation temperature of TiO_2_.

**Figure 2 nanomaterials-13-00132-f002:**
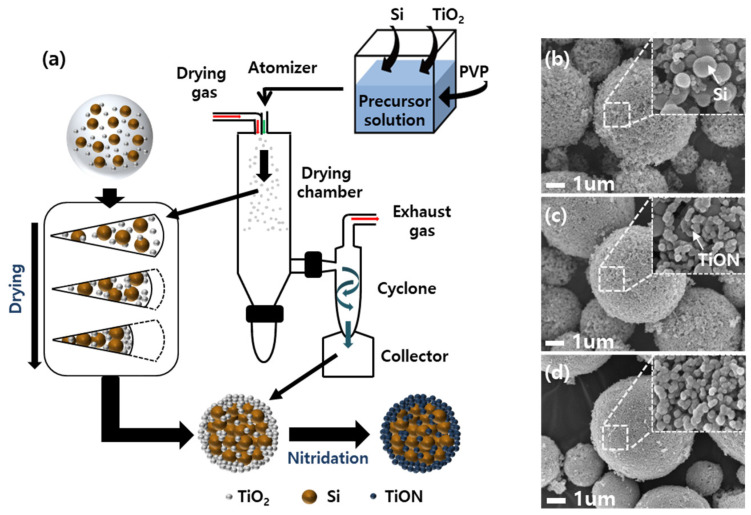
(**a**) Graphical illustration for synthesis of Si-TiON composite using spray-drying and subsequent nitridation. FE-SEM images of Si-TiON composite synthesized using different molar ratios of Si and TiON: (**b**) 1:0.5 (**c**) 1:1 (**d**) 1:2. Inset represents high-resolution images.

**Figure 3 nanomaterials-13-00132-f003:**
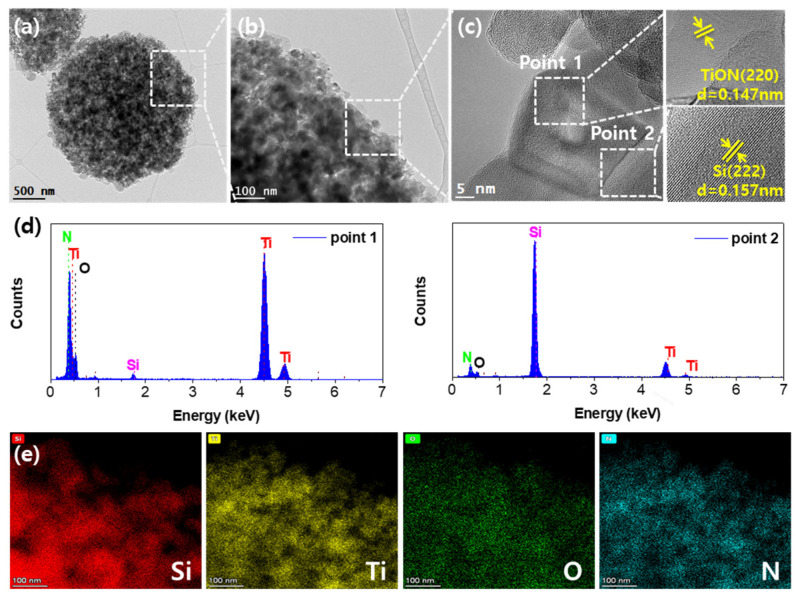
HRTEM images of Si-TiON (1:1) composite at (**a**) low and (**b**,**c**) high magnifications. Right inset shows lattice fringes within the TiON coating layer and Si core particle along directions (004) and (222), respectively. (**d**) Energy-dispersive X-ray spectra from points 1 and 2 in (**c**). (**e**) Si, Ti, O, and N elemental mapping indicating that TiON was well-distributed in the shell region.

**Figure 4 nanomaterials-13-00132-f004:**
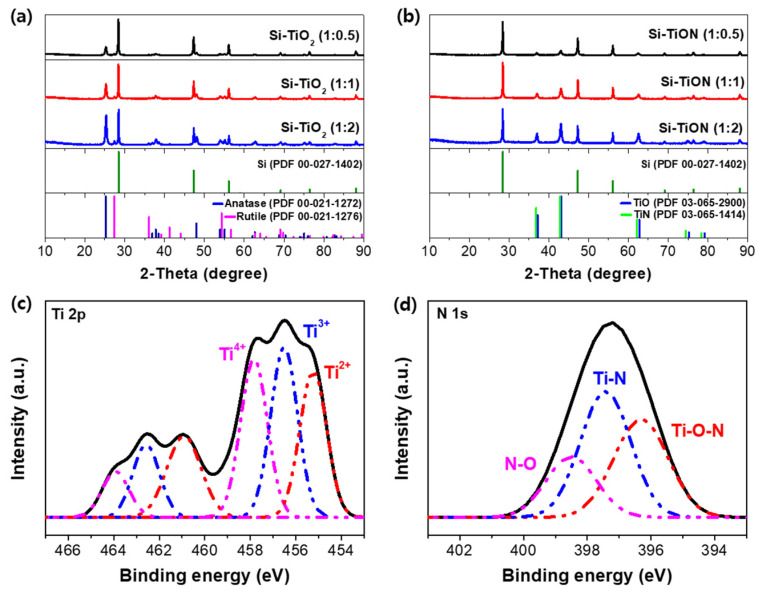
XRD patterns of (**a**) Si-TiO_2_ prepared by the spray-drying process and (**b**) subsequently nitrided Si-TiON composite with ratios of 1:0.5, 1:1, and 1:2. (**c**,**d**) X-ray photoelectron spectra for Si-TiON (1:1) composite in the region of binding energies of Ti 2p and N 1s core levels.

**Figure 5 nanomaterials-13-00132-f005:**
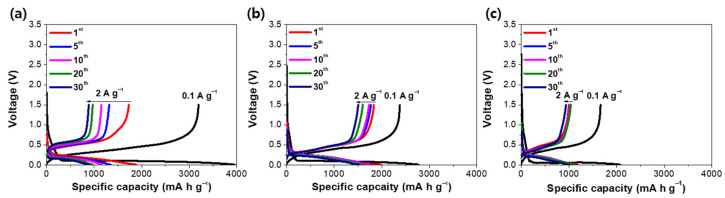
Voltage profiles for first and selected subsequent cycles at 0.1 A g^−1^ and 2 A g^−1^, respectively: (**a**) Si NPs, (**b**) Si-TiON (1:0.5), and (**c**) Si-TiON (1:1).

**Figure 6 nanomaterials-13-00132-f006:**
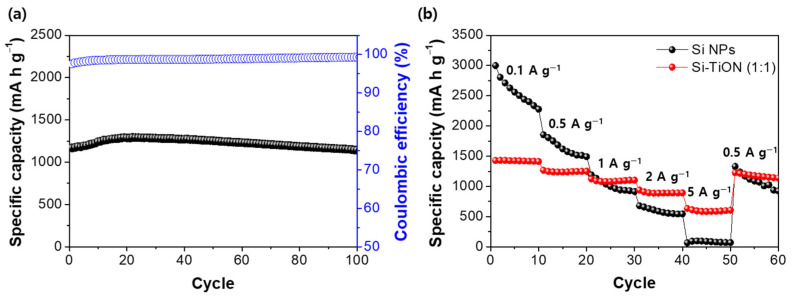
(**a**) Cycling performance of Si-TiON (1:1) measured at 1 A g^−1^ and (**b**) comparison of rate performances of Si NPs and Si-TiON (1:1) at different current densities.

## Data Availability

Data are contained within the article or Appendix A.

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
