# Peer review of "Achieving Cycling Stability in Anode of Lithium-Ion Batteries with Silicon-Embedded Titanium Oxynitride Microsphere"

_nanomaterials, 2022, doi:10.3390/nano13010132_

Round 1

Reviewer 1 Report

This manuscript reports an Si embedded titanium oxynitride (Si-TiON) obtained by using spray drying process. The TiON is uniformly coated on the Si surface by self-assembly, which improves the Si utilization and electrode stability. Si-TiON electrodes deliver initial reversible capacity of 1140 mA h g-1 and enhanced cyclic stability. In this work, some interesting and significant findings are reported. Thus, we can recommend its publication in Nanomaterials after major revision. Here are my detailed comments:

1. How much are the mass fractions of Si in Si-TiON (1:0.5), Si-TiON (1:1) and Si-TiON (1:2).

2. Please add a table to compare the eletrochemical performances of Si-TiON (1:1) and reported Si-based anodes in previous works.

3. To better evaluate the advantages of TiON coating, please provide the cyclic life of Si-TiON (1:2) and Si-TiO2 (1:1).

4. To prove the better stability of Si-TiON (1:1) than that of Si NPs, some analyses on electrode after cyclic test are important.

5. Some recent literatures about lithium ion batteries are suggested to help the authors get deeper insight into the characterization of material (10.1016/j.jcis.2021.10.113; 10.20517/energymater.2021.21; 10.20517/energymater.2021.15).

6. There are some minor mistakes in manuscript, please check manuscript carefully. For example, “0.1A g-1” should be “0.1 A g-1” in Fig. 6b.

Reviewer 2 Report

In the work, authors report the spray-dry method to synthesize the silicon embedded titanium oxynitride microspheres for lithium-ion batteries as anodes. The work is interesting, while for better readership, several related contributions should be enriched in the work for better understanding the work.

1.      TiON own high conductivity, however, for the case here, please give some specific values for the as-obtained TiON here.

2.      Li diffusion coefficients by using GITT or EIS should be provided in the revised version for better understanding the striking electrochemical Li-storage performance of the composite anode here.

3.      Mixed reference style can be found in the reference section, along with discernable spelling/grammar mistakes in the manuscript. Please check and revise them one by one carefully.

4.      Several related contributions (Adv. Energy & Sustain. Res. 2022, 3, 2200028; Adv. Energy Mater. 2021, 11, 2100287) should be enriched in the work for better understanding the work.

5.      As for the TiON phase, please add the standard JCPDF card No in the revised version to confirm the phase-pure TiON.

Round 2

Reviewer 1 Report

The manuscript has been revised well.